# Effect of Organo-Modified Montmorillonite Nanoclay on Mechanical, Thermo-Mechanical, and Thermal Properties of Carbon Fiber-Reinforced Phenolic Composites

**DOI:** 10.3390/polym13050754

**Published:** 2021-02-28

**Authors:** Jantrawan Pumchusak, Nonthawat Thajina, Watcharakorn Keawsujai, Pattarakamon Chaiwan

**Affiliations:** 1Department of Industrial Chemistry, Faculty of Science, Chiang Mai University, Chiang Mai 50200, Thailand; nonthawat_1234@hotmail.com (N.T.); few15857@gmail.com (W.K.); pattarakamon.name@gmail.com (P.C.); 2Center of Excellence in Materials Science and Technology, Chiang Mai University, Chiang Mai 50200, Thailand

**Keywords:** carbon fiber-reinforced polymer composite, phenolic resin, nanoclay, thermo-mechanical property, heat resistant composite

## Abstract

This work aims to explore the effect of organo-modified montmorillonite nanoclay (O-MMT) on the mechanical, thermo-mechanical, and thermal properties of carbon fiber-reinforced phenolic composites (CFRP). CFRP at variable O-MMT contents (from 0 to 2.5 wt%) were prepared. The addition of 1.5 wt% O-MMT was found to give the heat resistant polymer composite optimum properties. Compared to the CFRP, the CFRP with 1.5 wt% O-MMT provided a higher tensile strength of 64 MPa (+20%), higher impact strength of 49 kJ/m^2^ (+51%), but a little lower bending strength of 162 MPa (−1%). The composite showed a 64% higher storage modulus at 30 °C of 6.4 GPa. It also could reserve its high modulus up to 145 °C. Moreover, it had a higher heat deflection temperature of 152 °C (+1%) and a higher thermal degradation temperature of 630 °C. This composite could maintain its mechanical properties at high temperature and was a good candidate for heat resistant material.

## 1. Introduction

Carbon fiber-reinforced polymer composites (CFRP) have been widely used in a vast range of fields, including aerospace, defense, and automotive industries, because of their high specific strength, modulus, stiffness, good chemical resistance, and low density [1,2]. Carbon fibers are increasingly used in numerous applications due to their outstanding features, e.g., high tensile strength, high modulus, high thermal stability, excellent corrosion resistance, and light weight, as well as lower consumption cost [3,4]. Unfortunately, carbon fibers have an inevitable drawback resulting from their weak interfacial adhesion with most polymer matrices due to their smooth graphitic surfaces, chemical inertness, low surface energy, and stable non-polar structures, affecting the properties of polymer composites [5]. It is well known that the fiber–matrix interfacial interaction ensures the stress transfer from the weak matrix to the strong fiber. The absence of strong interfacial adhesion can constrain the performance of polymer composites and hinders the real potential of carbon fibers [6]. The improvements of the carbon fiber–polymer matrix interface are divided into two main approaches: matrix modification and fiber treatment [7,8,9,10].

Multi-scale carbon fiber-reinforced polymer composites containing microscale fibers as the primary reinforcement and nanoscale fillers such as nanosilica [11,12], carbon nanotube [13,14], graphene derivative [15,16], and nanoclay [17,18] as the secondary reinforcement have attracted great attention in diverse applications. High aspect ratio nano particles provide an extremely large interfacial area between the particles and the host material, leading to a strong reinforcement even at low filler content [19]. It is expected that the polymer matrix can be strengthened by the nanoscale fillers, resulting in the enhancement of the performance of the multi-scale carbon fiber-reinforced polymer composite. The modification of the composite interface was described from a variety of interface theories such, as chemical bonding, wetting, mechanical interlock, and local stiffness of polymer matrix [16].

Nowadays, attention on nanoclay–polymer composites has increased because of their outstanding characteristics [20]. Nanoclay–polymer composites can be divided into two categories: intercalated and exfoliated composites. In an intercalated composite, a few polymer chains diffuse into the silicate galleries with fixed interlayer spacing. In an exfoliated composite, the silicate nanolayers are delaminated and well dispersed in the polymer matrix. The interface between the organic and inorganic phases can be greatly increased in the exfoliated state, and the composites with optimum properties can then be obtained [18]. Montmorillonite nanoclay has been much considered to be the most attractive nanoclay as the secondary reinforcing material for carbon fiber-reinforced polymer composites due to its high aspect ratio (100–1000), high modulus (170 GPa), large surface area (750 m^2^/g), and high thermal properties [17,21,22]. It was reported that the addition of a small content of nanoclay particles could considerably improve the stiffness and strength of polymer composites. Most importantly, exfoliation of the nanoclay platelets in the polymer matrix is the key condition to get improved composite properties. Jahangiri et al. [11] reported that short carbon fiber–phenolic composites reinforced with 1.0 wt% of nanoclay particles had the highest tensile and flexural strength. Islam et al. [13] showed that carbon fiber fabric–epoxy composites modified with 2.0 wt% of montmorillonite nanoclay provided the highest flexural strength, flexural modulus, storage modulus, and glass transition temperature. Rao et al. [22] prepared carbon fiber fabric–phenolic composites with montmorillonite nanoclay as a filler and reported that the flexural strength, flexural modulus, and thermal stability of the composite increased when the nanoclay loading increased up to 2.0 wt% of nanoclay and decreased when the nanoclay loading was more than 2.0 wt%. Tareq et al. [23] exhibited the flexural strength and storage modulus improvements of carbon fiber fabric–epoxy composites by 9% and 25%, respectively, by the inclusion of 2.0 wt% montmorillonite nanoclay. M. Hosur et al. [24] showed that the addition of 2.0 wt% montmorillonite nanoclay improved the flexural strength, flexural modulus, and storage modulus of epoxy composites by 19%, 35%, and 12%, respectively. 

In our previous work, we studied the synergistic effects of multi-fillers on the mechanical properties of short carbon fiber-reinforced phenolic composites and demonstrated that the incorporation of nanosilica, carbon black, and poly(acrylonitrile-co-butadiene) rubber greatly increased mechanical and thermo-mechanical properties when compared to the neat phenolic resin [25]. However, the composite still had a low thermal property, which is one of the important properties for heat resistant materials (e.g., automotive part and aerospace applications). The preservation of high mechanical properties at high temperature is crucial in these applications and is very challenging to study. In this work, montmorillonite nanoclay was used to meet this goal. Nanoclay are generally treated with organic modifiers so as to improve the compatibility between the nanoclay and polymer matrix. To the best of the authors’ knowledge, there is no substantial work that has researched the effect of organo-modified montmorilonite nanoclay on the mechanical, thermo-mechanical, and thermal properties of short carbon fiber-reinforced phenolic composites. Thus, the objective of this work was to find the optimum content of organo-modified montmorilonite nanoclay as a secondary reinforcing filler in short carbon fiber-reinforced phenolic composites for enhancing their properties. 

## 2. Materials and Methods

### 2.1. Materials

Novolac-type phenolic resin powder (PGA-2470, Thai GCI Resitop Co., Ltd., Rayong, Thailand) was used as a polymer matrix. Chopped CF (TR066A, Mitsubishi Chemical Corporation, Tokyo, Japan) with a diameter of 7 µm and a length of 6 mm was used as a primary reinforcement. Nanoclay in this study (Nanomer® I.28E, Sigma-Aldrich, St. Louis, MO, USA) was surface-modified montmorillonite (O-MMT) by 25–30 wt% trimethyl stearyl ammonium. The bulk density of the clay was 0.2–0.5 g/cm^3^. Zinc stearate (Acros Organics, Belgium) was used as a lubricant. 

### 2.2. Preparation of the CFRP/O-MMT Composites

The CFRP, which consisted of 66.5–69.0 wt% of phenolic resin, 30.0 wt% of carbon fiber, and 1.0 wt% of zinc stearate, was prepared. Added into the CFRP to improve its mechanical, thermo-mechanical, and thermal properties were 0.5, 1.0, 1.5, 2.0, and 2.5 wt% of O-MMT. All components were mixed by dry ball milling for 3 h to get the homogeneous composite mixtures. After mixing, the mixtures were fabricated into specimens by hot compression molding at 150 °C and 12.4 MPa and then the specimens were post-cured in an oven at 150 °C for 1 h and 180 °C for another 2 h.

### 2.3. Characterization

Density of the polymer composites was measured according to ASTM D792 using the Archimedes method. The average data of 3 replicate specimens were recorded.

The void content in the polymer composites was calculated according to Equation (1) as per ASTM D2734-94.
(1)$$Void\;content=\;\left(\frac{\rho_{\mathrm{theoretical}}-\;\rho_{\mathrm{experimental}}}{\rho_{\mathrm{theoretical}}}\right)\times 100\%$$

The theoretical density of the polymer composites according to weight fraction and density of each fiber or filler component was determined according to Equation (2).
(2)$$\rho_{\mathrm{theoretical}}\;\left(g/cm^{3}\right)=\;\frac{1}{\left(\frac{W_{\mathrm{PF}}}{\rho_{\mathrm{PF}}}+\frac{W_{\mathrm{CF}}}{\rho_{\mathrm{CF}}}+\frac{W_{\mathrm{ZnSt}}}{\rho_{\mathrm{ZnSt}}}+\frac{W_{O-\mathrm{MMT}}}{\rho_{O-\mathrm{MMT}}}\right)}$$
where $W_{\mathrm{PF}},\;W_{\mathrm{CF}},\;W_{\mathrm{ZnSt}},\;W_{O-\mathrm{MMT}}$ are the weight fraction of phenolic resin, carbon fiber, zinc stearate, and organo-modified montmorillonite nanoclay, respectively, while $\rho_{\mathrm{PF}},\;\rho_{\mathrm{CF}},\;\rho_{\mathrm{ZnSt}},\rho_{O-\mathrm{MMT}}$ are the densities of phenolic resin, carbon fiber, zinc stearate, and organo-modified montmorillonite nanoclay, respectively.

Tensile tests were performed on a Lloyd universal testing machine (model LRX, Largo, FL, USA) according to ASTM D638, with a crosshead speed of 2.0 mm/min. Three-point bending tests were performed on the same machine according to ASTM D790 with a crosshead speed of 2.0 mm/min. Charpy impact tests were measured by Jinan testing equipment (TEJC-50, China) at room temperature according to ISO179. Five samples were tested in each test. The average value and standard deviation were calculated for each composite. 

Dynamic mechanical analysis (DMA) was performed on a DMA1 (Mettler-Toledo, Columbus, OH, USA) equipped with single cantilever clamp and sample size of 10.0 mm × 5.0 mm × 1.0 mm. The analysis was carried out from 30 to 300 °C at a heating rate of 5 °C/min by using a constant frequency of 1 Hz. 

The heat deflection temperature was determined by a Ceast-6970 heat deflection temperature tester (Norwood, MA, USA). Samples having a size of 80.0 mm × 10.0 mm × 4.0 mm were collected for testing.

Thermal stability was determined by a Rikagu TG-DTA8120, Japan. Approximately 10 mg of each sample was placed on an alumina pan for testing. The measurement was performed from 25 to 800 °C at a heating rate of 10 °C under N_2_ atmosphere with a flow rate of 80 mL/min. 

A field-emission scanning electron microscope (FE-SEM, JEOL JSM-6335 F, Japan) was used to observe the fracture surface morphology of the polymer composites. 

## 3. Results

### 3.1. Density and Void Content of the CFRP/O-MMT Composites

Table 1 shows the theoretical density, experimental density, and void content of the CFRP/O-MMT composites. The density of O-MMT, carbon fiber, and phenolic resin used in this work were 0.2–0.5 g/cm^3^, 1.8 g/cm^3^, and 1.3 g/cm^3^, respectively. Although, the density of O-MMT was lower than that of the neat phenolic resin, it was found that with the increase of O-MMT content in the CFRP, the experimental density slightly increased. This can be ascribed to the nanoclay particles that filled up the air gap or void between the reinforcing carbon fiber and phenolic matrix [26]. However, non-uniformity was found in the composites. Note that the non-uniform material presented on the fracture surface, which can be seen in the SEM image (Figure 8), interfered in the relationship between uniform physical value and filler content. The experimental density of the CFRP and CFRP with 0.5 and 1.0 wt% were lower than those of the theoretical density because of the void formation during the fabrication of the polymer composites. It can be seen that as the O-MMT loading in the CFRP increased, the void contents of the CFRP were coming down. This corresponds to SEM image in which the composites added higher contents of O-MMT and showed more dense materials and lower presented voids. The CFRP with 1.5, 2.0, and 2.5 wt% O-MMT had no void content among all CFRP. This is due to a better interaction or adhesion bonding of the surface modified MMT, with better wetting surface capability. It is noted that the theoretical density of the composites showed the opposite tendency, which is due to the void formation in the composites that had not been considered for the theoretical density calculation. So, the theoretical density of the composites tended to decrease as the O-MMT content increased.

### 3.2. Mechanical Properties of the CFRP/O-MMT Composites

Figure 1 shows the tensile strength and Young’s modulus of the CFRP/O-MMT composites. With the incorporation of O-MMT, the tensile strength of the CFRP increased and reached 64.3 ± 4.6 MPa at 1.5 wt% O-MMT, which was 20% higher than that of the CFRP (53.7 ± 13.2 MPa). More O-MMT loading showed no further increase in the tensile strength because of the agglomeration of O-MMT, as shown by the FE-SEM observation. Besides, the CFRP with all O-MMT contents possessed similar Young’s modulus at 1.0 ± 0.1 GPa, which were 11% higher than that of the CFRP (0.9 ± 0.1 GPa). Jahangiri et al. [11] also achieved similar results, and they stated that the two main reasons for strengthening the polymer matrix with nanoclay particles were the filling of void volumes and the strength of composite reinforced with nanoclay particles.

Figure 2 shows the bending strength and bending modulus of the CFRP/O-MMT composites. The bending strength of the CFRP decreased at a low O-MMT content of 0.5 wt%. As the O-MMT loading increased, the bending strength of the CFRP became higher. The CFRP with 2.0 wt% O-MMT gave the highest bending strength of the CFRP (189.0 ± 32.0 MPa), which improved by 15% compared to the CFRP (164.3 ± 28.0 MPa). The bending strength of the CFRP marginally decreased at a higher O-MMT loading. It is noted that the bending strength of the composites can be enhanced by adding O-MMT; however, the relationship between the bending strength and O-MMT content are interfered by the non-uniformity of the composites. Moreover, the bending modulus of the CFRP increased when the O-MMT loading increased. The addition of 2.0 wt% O-MMT offered the highest bending modulus (11.2 ± 0.8 GPa), which improved by 11%, compared to the CFRP (10.1 ± 0.5 GPa). At a high O-MMT content of 2.5 wt%, the bending modulus of the CFRP reduced to some extent. This can be attributed to the enhancement of the chemical bonding between the polymer matrix and the organically modified nanoclay, strengthening the matrix further. However, at higher nanoclay loading, the mechanical locking descended, which caused the decrease in strength and modulus [22].

Impact resistance is a vital property for all materials, which is the ability of the materials to resist the fracture under sudden impact load [20]. Figure 3 shows the impact strength of the CFRP/O-MMT composites. It was found that the CFRP with 0.5 wt% O-MMT showed the highest impact strength (55.5 ± 20.6 kJ/m^2^), which improved by 73%, compared to the CFRP (32.2 ± 9.9 kJ/m^2^). This is explained by the finding that the impact strength of the composites can be efficiently improved by adding the small content of O-MMT as 0.5 wt%, which is different from the results of the tensile and bending testing. Thus, the materials may have responded differently to the different mode of loading. The 0.0% O-MMT is the CFRP composite; it showed its own properties, but once 0.5% O-MMT was added because of its good distribution in the CFRP composite, it could provide the effective toughening mechanism to the composite and higher impact strength was obtained. The good distribution is shown in Figure 8a–c. In Figure 8b, the step structure surface was observed in the matrix phase, which indicates that it is the more ductile characteristic or the tougher material. When the O-MMT loading increased, the impact strength of the CFRP diminished but were still greater than that of the CFRP. This is due to the O-MMT agglomeration at high adding contents. Similar improvement of the impact strength in the nanoclay–polymer composites was also reported recently [27]. 

The mechanical properties were even better than those of the CFRP reinforced with nanosilica/carbon black/poly(acrylonitrile-co-butadiene) rubber in our previous work [25], confirming that the O-MMT can greatly reinforce the phenolic matrix as a consequence of homogeneous dispersion and strong interfacial adhesion between the O-MMT particles and the phenolic matrix.

### 3.3. Thermo-Mechanical Properties of the CFRP/O-MMT Composites

A thermo-mechanical analysis was carried out by DMA to investigate the viscoelastic properties of the polymer composites. The changes in storage modulus (*E*‘), loss modulus (*E*“), and tan delta (δ) of the CFRP/O-MMT composites with response to temperature are presented in Figure 4. The DMA results of the CFRP/O-MMT composites are summarized in Table 2. As can be seen, the introduction of O-MMTs significantly increased the storage modulus considerably in both the glassy and rubbery region in comparison to the CFRP. An increase in storage modulus with the increasing O-MMT loading was seen. For the CFRP with 2.5 wt% O-MMT, the storage modulus at 30 °C yielded the highest value of 6.5 GPa, which was 67% greater than that of the CFRP. It also could reserve its high modulus up to 145 °C. The improvement in the storage modulus can be attributed to mechanical and chemical interlocking due to the addition of nanoparticles that restricted polymer chain mobility. Figure 5 shows the chemical structure of trimethyl stearyl ammonium modified MMT. As we can see, the ammonium ion can bond with the hydroxy methyl group in the phenolic resin. Thus, the interfacial adhesion between O-MMT and phenolic matrix were increased, leading to higher stiffness of the polymer composite.

Loss modulus is regarded as a material’s ability to dissipate energy into heat when deformed under load. It was seen that the CFRP/O-MMT composites exhibited a more enhanced loss modulus than the CFRP. This can be ascribed to the improved fiber-polymer matrix interfacial bonding accomplished by the addition of O-MMT that improved shear stress and energy dissipation of the composites and thus increased the overall loss modulus [23,24].

In a DMA test, tan δ is the important parameter, which is defined as a ratio of lost energy to stored energy and is called a damping factor. The glass transition temperature (*T*_g_) was assigned as the peak position of tan δ curves. The incorporation of O-MMT increased the *T*_g_ of the CFRP; however, the CFRP with 0.5 wt% O-MMT showed a little decrease in *T*_g_. The increase in *T*_g_ is due to the restricted mobility of the polymer chains caused by the strong interactions between nanoparticles and the polymer matrix. Besides, it was observed that the tan δ peak height decreased when the addition of O-MMT increased, except for the CFRP, which suggested a reduction in the degree of molecular mobility. In the CFRP, the peak of the tan δ curve versus temperature was broadened. Allahverdi et al. [28] and Etemadi et al. [29] explained that a broadened tan δ curve resulted from the heterogeneity increase of the crosslinked structure of the CFRP. 

In order to get a better understanding about the effect of O-MMT on the curing behavior of the CFRP, the crosslink density was calculated using rubber elasticity theory based on the elastic modulus of samples at rubbery state obtained from DMA as described below [29,30]: (3)$$V_{e}=\;\frac{E^{\prime }}{3RT}$$
where $V_{e}$ is the crosslink density expressed in moles of elastically effective network chains per cubic centimeter of sample, *E*´ is the storage modulus in the rubbery region at a temperature well above *T*_g_, *R* is the universal gas constant, and *T* is the absolute temperature at which experimental modulus was selected (in this study *T* = 493.15 K). It can be seen that the addition of O-MMT into the CFRP increased the crosslink density (Table 2). In the CFRP/O-MMT composites, the improvement of crosslink density was consistent with the reduction in tan δ peak height and increase in *T*_g_. This can be ascribed to the participation of O-MMT in the curing reaction of phenolic resin, resulting in strong interfacial adhesion between the inorganic phase and the polymer matrix and the improved crosslinking density of the composites [21].

### 3.4. Thermal Properties of the CFRP/O-MMT Composites

The thermal properties of the CFRP/O-MMT composites were described by heat deflection temperature, which is the temperature at which a polymeric material deforms under a specified load. Figure 6 shows the heat deflection temperature of the CFRP/O-MMT composites. By increasing the O-MMT loading, the heat deflection temperature increased and reached 152.4 ± 1.8 °C at 1.5 wt% O-MMT, which was 1% higher than that of the CFRP (150.9 ± 2.3 °C). As the O-MMT loading further increased, the heat deflection temperature of the CFRP substantially decreased because of its agglomeration. This can be explained by the finding that that the nanoclay had a good thermal resistance property, thus it could improve the thermal properties of the composites. However, the nanoclay could agglomerate at a high adding content, which could possibly cause the formation of the phase of polymer and filler enrichment, as shown in Figure 8i.

### 3.5. Thermal Stability of the CFRP/O-MMT Composites

The thermal stability of the O-MMT and the CFRP/O-MMT composites were examined by TGA under N_2_ atmosphere at a heating rate of 10 °C/min. The TG and DTG curves are shown in Figure 7. The onset degradation temperature (*T*_5%_), maximum degradation temperature (*T*_dmax_), and the char residual weight at 800 °C (*R*_800_) are also shown in Table 3. The TGA results show that the thermal degradation of O-MMT occurred in the range of 200–450 °C. This weight loss could be attributed to the degradation of the organo-modifier part of nanoclay. The residual weight of clay at 800 °C was about 65%. The *T*_5%_ of the CFRP/O-MMT composites were around 380–392 °C, attributed to the degradation of methylene bridges in the phenolic resin. As the temperature increased, the TG curve of the CFRP/O-MMT composite showed a sharp drop, and the DTG curve showed the highest weight loss rate, corresponding to degradation of the alkyl chain. When the temperature increased above 600 °C, the weight loss rate of each composite was significantly decreased. It was indicated that most of the groups in the phenolic resin structure were thermally decomposed [30]. Moreover, the results show that the addition of O-MMT at a critical weight percentage of 1.5 wt% can increase the *T*_5%_, *T*_dmax_, and *R*_800_ of the CFRP composites to 392 °C, 630 °C, and 23%, respectively, which indicate that the thermal stability of the composite was increased, attributed to the good thermal resistance of nanoclay and the strong interaction nanoclay layers with the phenolic matrix; thus, it can hinder the interaction of oxygen and polymer matrix. It is noted that the degradation peak of the organo-modifier of MMT at 200–450 °C does not show in the TGA thermogram of composites. It means there is the binding between the O-MMT and phenolic matrix. 

### 3.6. Fracture Surface Morphology of the CFRP/O-MMT Composites

Figure 8 shows the fracture surface morphology of the CFRP/O-MMT composites. As can be seen in Figure 8a, the fracture surface of the CRFP was quite smooth. The pullouts of carbon fibers with smooth fiber surface were clearly observed, indicating poor fiber–matrix interfacial bonding and relatively brittle nature of the polymer matrix. Besides, the porosity was found in the CFRP as well, causing premature damage or failure in the composite. With O-MMTs addition (Figure 8b–j), the fracture surface of the CFRP became rougher, suggesting increased resistance of crack propagation in the composites [28]. In addition, a considerable amount of polymer matrix was stuck on the carbon fiber surface. This can be attributed to the addition of O-MMT, which can greatly improve the interfacial adhesion between the carbon fiber and phenolic matrix and which may be restricted fiber–matrix debonding and fiber pullouts. However, at the high O-MMT content, the agglomeration of O-MMT was observed, resulting in stress raisers that acted as a crack initiation site and led to premature failure, thus diminishing matrix and interfacial properties [23].

## 4. Conclusions

This work brings about the development of short carbon fiber-reinforced phenolic composite by incorporating organo-modified montmorillonite nanoclay and the investigation of its mechanical, thermo-mechanical, and thermal properties. The composites were prepared by adding 0.5–2.5 wt% O-MMT to the CFRP via dry ball milling and fabricated by hot compressing molding. The results obtained in this work are summarized as below.

The addition of O-MMT can efficiently enhance the interfacial interaction between carbon fiber and phenolic matrix, leading to the improvement of the CFRP.The use of 1.5 wt% O-MMT as a secondary reinforcement was optimum in the CFRP for improving by 20% tensile strength, 11% Young’s modulus, 4% bending modulus, 51% impact strength, 67% storage modulus, similar glass transition temperature, and 1% heat deflection temperature, compared to the CFRP. It increased the thermal degradation temperature of the CFRP to 630 °C. However, it showed a little lower bending strength, which decreased by 1%.Morphology analysis indicated that O-MMT significantly improved carbon fiber–polymer matrix interfacial bonding; nevertheless, O-MMT was found to agglomerate at a high loading content.Overall, it can be concluded that the incorporation of a small content of O-MMT is beneficial to the mechanical, thermo-mechanical, and thermal properties of CFRP for use as a heat resistant material.

## Figures and Tables

**Figure 1 polymers-13-00754-f001:**
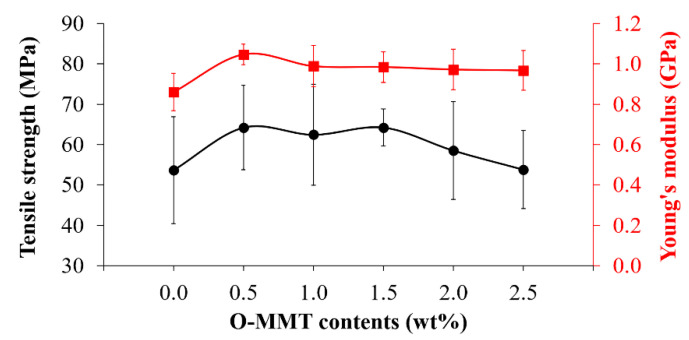
Tensile strength and Young’s modulus of the CFRP/O-MMT composites.

**Figure 2 polymers-13-00754-f002:**
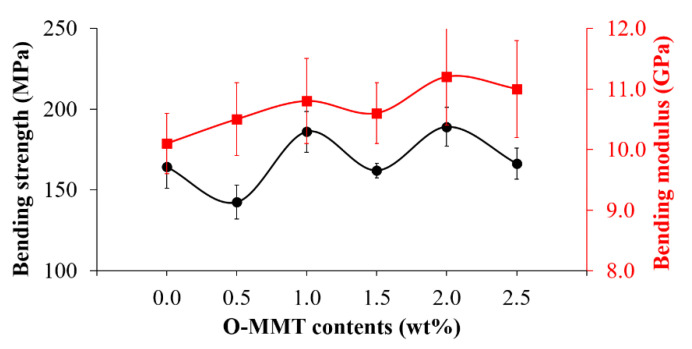
Bending strength and bending modulus of the CFRP/O-MMT composites.

**Figure 3 polymers-13-00754-f003:**
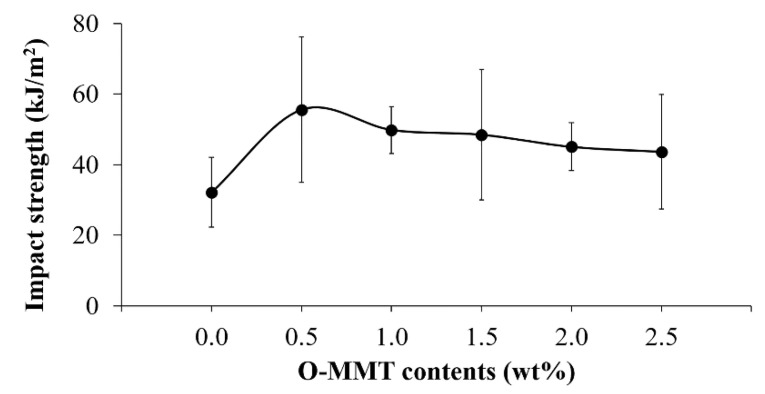
Impact strength of the CFRP/O-MMT composites.

**Figure 4 polymers-13-00754-f004:**
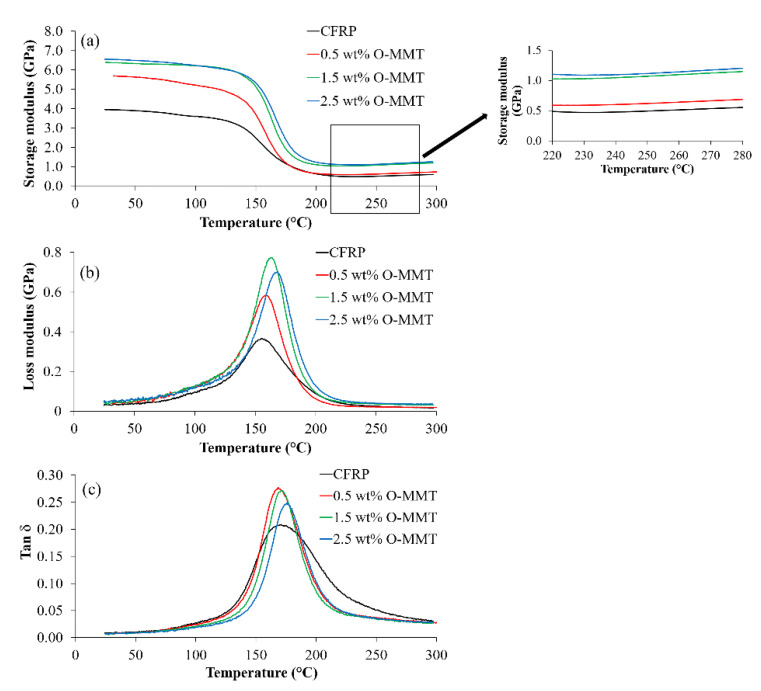
Changes in (**a**) storage modulus, (**b**) loss modulus, and (**c**) tan δ of the CFRP/O-MMT composites with response to temperature.

**Figure 5 polymers-13-00754-f005:**
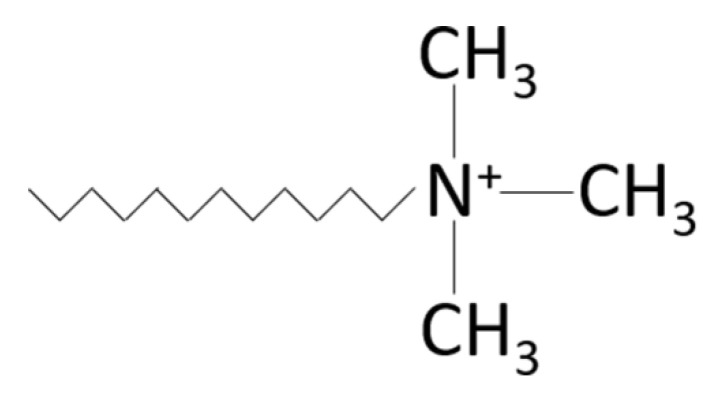
The chemical structure of trimethyl stearyl ammonium modified MMT.

**Figure 6 polymers-13-00754-f006:**
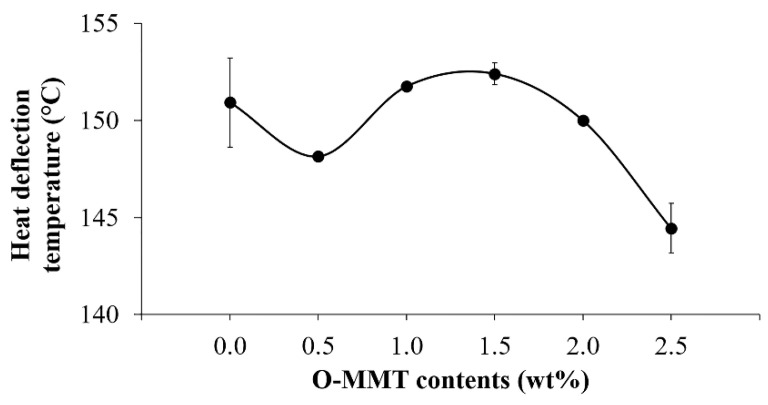
Heat deflection temperature of the CFRP/O-MMT composites.

**Figure 7 polymers-13-00754-f007:**
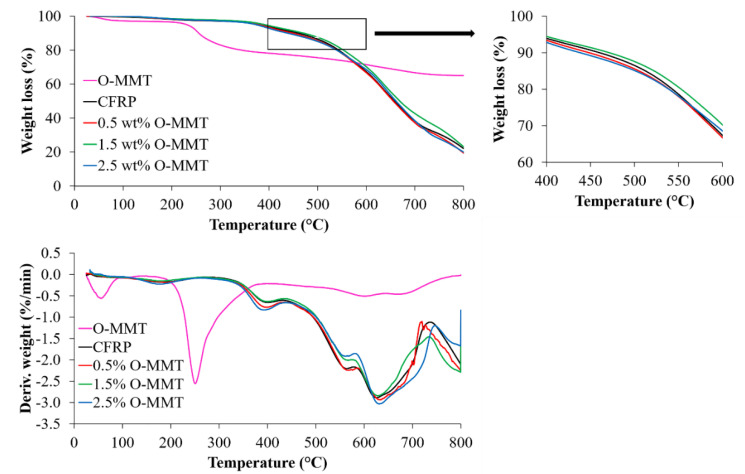
TG-DTG curves of the CFRP/O-MMT composites.

**Figure 8 polymers-13-00754-f008:**
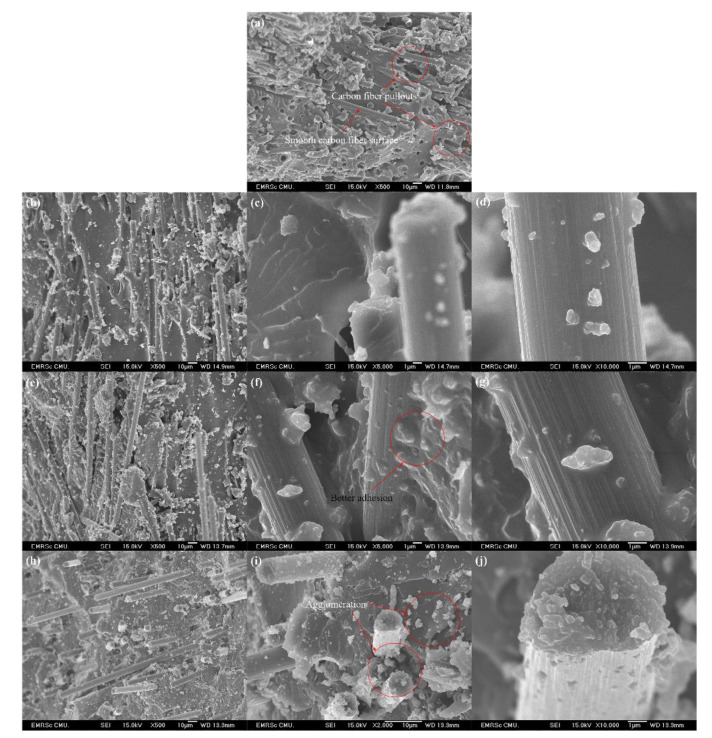
Fracture surface morphology of (**a**) the CFRP and the CFRP with O-MMT of 0.5 wt% (**b**–**d**), 1.5 wt% (**e**–**g**), and 2.5 wt% (**h**–**j**).

**Table 1 polymers-13-00754-t001:** Theoretical density, experimental density, and void contents of the carbon fiber-reinforced polymer composites (CFRP)/organo-modified montmorillonite (O-MMT) composites.

Composites	Experimental Density (g/cm^3^)	Theoretical Density (g/cm^3^)	Void Content (%)
CFRP	1.34 ± 0.02	1.42	5.6
0.5 wt% O-MMT	1.30 ± 0.03	1.40	7.1
1.0 wt% O-MMT	1.36 ± 0.03	1.38	1.4
1.5 wt% O-MMT	1.37 ± 0.02	1.36	-0.7
2.0 wt% O-MMT	1.36 ± 0.01	1.34	-1.5
2.5 wt% O-MMT	1.35 ± 0.01	1.32	-2.3

**Table 2 polymers-13-00754-t002:** Dynamic mechanical analysis (DMA) results of the CFRP/O-MMT composites.

Composites	*E*‘ at 30 °C (GPa)	% Change	*E*“at Peak (GPa)	% Change	*T*_g_ (°C)	% Change	$V_{e}\;(\mathrm{mol}/{\mathrm{cm}}^{3})$
CFRP	3.9	-	0.4	-	170.6	-	0.04
0.5 wt% O-MMT	5.7	46.1	0.6	50.0	168.4	−1.3	0.05
1.5 wt% O-MMT	6.4	64.1	0.8	100.0	170.9	0.2	0.08
2.5 wt% O-MMT	6.5	66.7	0.7	75.0	175.5	2.9	0.09

**Table 3 polymers-13-00754-t003:** TGA data of the CFRP/O-MMT composites.

Composites	*T*_5%_^1^ (°C)	*T*_dmax_^2^ (°C)	*R*_800_^3^ (wt%)
O-MMT	231.6	251.0	65.1
CFRP	383.5	623.5	22.0
0.5 wt% O-MMT	379.8	632.1	19.5
1.5 wt% O-MMT	392.1	630.0	23.3
2.5 wt% O-MMT	373.8	633.1	20.0

^1^ Onset degradation temperature, ^2^ Maximum degradation temperature, ^3^ Char residual weight at 800 °C.

## Data Availability

Not applicable.

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
