# Peer review of "Effect of Organo-Modified Montmorillonite Nanoclay on Mechanical, Thermo-Mechanical, and Thermal Properties of Carbon Fiber-Reinforced Phenolic Composites"

_polymers, 2021, doi:10.3390/polym13050754_

Round 1

Reviewer 1 Report

This paper reports on preparation, characterization and mechanical properties of O-MMT CFPR materials.

From the view point of application, too, this study is worth publishing in Polymers essentially.

Before acceptance, some difficult points should be explained closely.

(1)Figure 2

Why black points (lines) exhibited increasing and decreasing (not linear to O-MMT contents)? Please explain the reason in the text more easily.

(2) Figure 3

Why 0.5 exhibited the highest values (not linear to O-MMT contents, otherwise 0.0 exhibited lower value than 0.5)?  Please explain the reason in the text more easily.

(3) Figure 5

Similar to Figure 3, Figure 5 also exhibited the highest value at 0.5 (not 0.0). Please explain the reason in the text more easily and mention the similarity to Figure 3.

(4) Non-uniformity

The values ​​shown in Table 1 are "average" values, aren't they? Also mention the possibility that the non-uniform material present on the surface, such as the SEM image in Figure 6, interferes with the relationship between uniform physical value and content.

That's all.

Author Response

This paper reports on preparation, characterization and mechanical properties of O-MMT CFPR materials.

From the view point of application, too, this study is worth publishing in Polymers essentially.

  • We would like to thank you for your time and precious comments that make this manuscript highly qualified in the field of inorganic-polymer composites.

Before acceptance, some difficult points should be explained closely.

(1) Figure 2

Why black points (lines) exhibited increasing and decreasing (not linear to O-MMT contents)? Please explain the reason in the text more easily.

  • This is due to the non-uniformity of the composites as you give the comments in (4). We have explained the reason in Section 3.2, Line 200-204.

(2) Figure 3

Why 0.5 exhibited the highest values (not linear to O-MMT contents, otherwise 0.0 exhibited lower value than 0.5)?  Please explain the reason in the text more easily.

  • The 0.0% O-MMT is the CFRP composite, it showed its own properties, but once 0.5% OMMT was added because of its good distribution in the CFRP composite, so it could provide the effective toughening mechanism to the composite and higher impact strength was obtained. The good distribution is shown in Fig 7 (a, b, and c). In Fig.7 b, the step structure surface was observed on the matrix phase, this indicates that it is the more ductile characteristic or the tougher material. The impact strength of the composites decreased at high O-MMT adding contents due to its agglomeration. We have shown more explanations in Section 3.2, Line 218- 229.

(3) Figure 5

Similar to Figure 3, Figure 5 also exhibited the highest value at 0.5 (not 0.0). Please explain the reason in the text more easily and mention the similarity to Figure 3.

  • The nanoclay has a good thermal resistance property, thus it can improve the thermal properties of the composites. However, the nanoclay can agglomerate at a high adding content, which possibly cause the formation of the phase of polymer and filler rich as shown in Figure 7(i). We have explained the reason in Section 34, Line 295-299.

(4) Non-uniformity

The values ​​shown in Table 1 are "average" values, aren't they? Also mention the possibility that the non-uniform material present on the surface, such as the SEM image in Figure 6, interferes with the relationship between uniform physical value and content.

That's all.

  • Yes, the density values shown in Table 1 are average values. We measured the density of 5 replicate specimens and showed the average values of them. We agreed with your comments and added more discussions in Section 3.1, Line 159-162. The dispersion of nano fillers in the multi-scale polymer composite is very challenge. The non-uniformity can be found in the composites. Nanoparticle dispersion in polymer matrix needs special mixing method, such as ultrasonication; however, this is not suitable for the real industrial production. In this study, the ease of production process is concern.

Reviewer 2 Report

In this contribution the authors investigated the preparation and properties of organo modified montmorillonite addition to carbon fiber reinforced phenolic composite. The composite shows higher tensile strength and impact strength in comparison with unmodified composite.

CF shortcut is not explained. It should be better to use Pa instead of psi. Are the authors sure that all volatile compounds were removed from composite? Maybe, thermogravimetric study should be applied. Is there any explanation for curve of bending strength in Fig. 2? What happens with the composites at temperatures above 150 ℃? Was thermal stability of organo modified montmorillonite studied? Should not it be better to compare with montmorillonite alone?

Author Response

In this contribution the authors investigated the preparation and properties of organo modified montmorillonite addition to carbon fiber reinforced phenolic composite. The composite shows higher tensile strength and impact strength in comparison with unmodified composite.

  • We would like to thank you for your time and precious comments that make this manuscript highly qualified in the field of inorganic-polymer composites.

CF shortcut is not explained.

  • In this work, we used the commercial chopped-CF from Mitsubishi Chemical Corporation. The diameter and length of CF have shown in Section 2.1, Line 99-100.

It should be better to use Pa instead of psi.

  • We have changed the pressure value from 1800 psi to 12.4 MPa in Section 2.2, Line 111.

Are the authors sure that all volatile compounds were removed from composite? Maybe, thermogravimetric study should be applied.

  • We applied the thermogravimetric study to the composites and added the result that shown in Section 3.5, Line 302-326.

Is there any explanation for curve of bending strength in Fig. 2?

  • We have explained the reason of the bending strength curve in Section 3.2, Line 201-204. The dispersion of nano fillers in the multi-scale polymer composite is very challenge. The non-uniformity can be found in the composites. Nanoparticle dispersion in polymer matrix needs special mixing method, such as ultrasonication; however, this is not suitable for the real industrial production. In this study, the ease of production process is concern.

What happens with the composites at temperatures above 150 ℃?   

  • The heat deflection temperature of composites is in the range of 150 °C (see Fig. 5). Therefore, at above 150 °C, composite can be deformed by loading more easily than the lower temperature.
  • In Fig. 4, it can be seen that the modulus of composite has dropped at above 150 °C since it approaches the Tg of composite (about 168-175 oC)
  • TGA results showed the thermal degradation steps of the composites which occurred at temperatures far above 150 °C (Figure 6 and Table 3).

Was thermal stability of organo modified montmorillonite studied? Should not it be better to compare with montmorillonite alone?

  • Yes, the thermal stability of O-MMT was studied and shown in Section 3.5, Line 302-326. However, the study of the montmorillonite alone does not include in this manuscript, but was published elsewhere. We used the organo modified montmorillonite because we want to improve the compatibility of the clay and polymer matrix.

Reviewer 3 Report

The presented article discusses the subject of the application of MMT particles in the modification of CF reinforced composites based on phenolic resin. In my opinion, the subject of the works is not innovative, the concept of producing this type of hybrid materials has been developed for many years. The work does not present new methods of processing this type of materials, nor does it discuss any new results or research methods. In the current form of works, I would recommend preparing a slightly different, more complex concept of works and paper resubmission. Below are some detailed comments on the presented results.

1) The section on density measurements is the weakest element of the work, the authors claim that MMT particles are able to fill the pores formed in the composite, while the majority of studies indicate the opposite tendency.

The authors nowhere indicate the actual density of MMT particles, but only the bulk density. Usually the density of MMT particles ranges from 1.9 to 2.7 g/cm3, which is a value higher than the density of carbon fibers at 1.8 g/cm3

In terms of the methodology of the measurement itself, I suggests using a helium pycnometer to measure the density of the input materials and the immersion method for already prepared samples.

2) The authors do not indicate what is the main reason for the significant increase in sample stiffness during DMTA measurements, in the case of static measurements the increase was not so large

3) Cross-link density analysis for this kind of composites is pointless, this type of comparison can be used for unfilled resins, while the use presented calculation for the presented comparison is rather confusing

Author Response

The presented article discusses the subject of the application of MMT particles in the modification of CF reinforced composites based on phenolic resin. In my opinion, the subject of the works is not innovative, the concept of producing this type of hybrid materials has been developed for many years. The work does not present new methods of processing this type of materials, nor does it discuss any new results or research methods. In the current form of works, I would recommend preparing a slightly different, more complex concept of works and paper resubmission. Below are some detailed comments on the presented results.

  • We would like to thank you for your time and precious comments that make this manuscript highly qualified in the field of inorganic-polymer composites.

1) The section on density measurements is the weakest element of the work, the authors claim that MMT particles are able to fill the pores formed in the composite, while the majority of studies indicate the opposite tendency.

  • In this work, we showed two types of density (experimental density and theoretical density). The experimental density tended to increase when the MMT adding contents increased. This claim that the MMT particles are able to fill the pores formed in the composite (the pores that usually occur when carbon composites are formed). This corresponds to SEM image that the composites added higher contents of O-MMT show more dense materials and lower presented voids. This result corresponds well to the work of Chee et.al. [ ref.26.Chee, S.S., Jawaid, M., Sultan, M.T.H., Alothman, O.Y., Abdullah, L.C. Effects of nanoclay on physical and dimensional sta-bility of Bamboo/Kenaf/nanoclay reinforced epoxy hybrid nanocomposites. J. Mater. Res. Technol. 2020, 9, 5871–5880; doi: 10.1016/j.jmrt.2020.03.114.] However, the theoretical density that we have calculated showed the opposite tendency. It is due to the voids formed in the composites have not been considered for the theoretical density calculation. So, the theoretical density of the composites tended to decrease as the O-MMT content increased. We have written more explanation in Section 3.1, Line 164-166 and 168-171.

The authors nowhere indicate the actual density of MMT particles, but only the bulk density. Usually the density of MMT particles ranges from 1.9 to 2.7 g/cm3, which is a value higher than the density of carbon fibers at 1.8 g/cm3

  • We have shown the actual density of MMT particles and carbon fibers as you recommend in Section 3.1, Line 154-155. However, we have shown that the density of commercial O-MMT is in the range of 0.2-0.5 g/cm3 which provided by Sigma Aldrich.

In terms of the methodology of the measurement itself, I suggests using a helium pycnometer to measure the density of the input materials and the immersion method for already prepared samples.

  • We would like to thank you for you recommend of using a helium pycnometer. However, we do not have this instrument in our area. So, we have used the density that provided by the company, which have the high standard measurement for this material. We have used the O-MMT from Sigma Aldrich and carbon fiber from Mitsubishi Chemical Corporation. The density measurement following ASTM D1895 that we have used in this work is typically used for engineering materials.

2) The authors do not indicate what is the main reason for the significant increase in sample stiffness during DMTA measurements, in the case of static measurements the increase was not so large

  • The results of the static and dynamic measurements could not compare directly, because the types of loading are different; therefore, the way the sample response can be different. For example, foam can be plastic in compression but brittle in tension.

3) Cross-link density analysis for this kind of composites is pointless, this type of comparison can be used for unfilled resins, while the use presented calculation for the presented comparison is rather confusing

  • Cross-link density of thermosetting polymer is important and relates to its mechanical property. We found that the crosslink density analysis of the polymer composites corresponds to the work of Ahmadijokani, F., Shojaei, A., Arjmand, M., Alaei, Y., Yan, N. Effect of short carbon fiber on thermal, mechanical and tribological behavior of phenolic-based brake friction materials. Compos. Part B 2019, 168, 98–105; doi:10.1016/j.compositesb.2018.12.038. They showed that the crosslink density value was used to confirm the improvement of thermo-mechanical property of polymer composites.

Round 2

Reviewer 2 Report

Thank you for the changes in manuscript.

Author Response

We would like to thank you again for your time and precious comments.

Reviewer 3 Report

According to the Authors' answers, all corrections were taken into account, which is not true. The authors only partially took into account the comments contained in the review. Until they are not included, I cannot recommend the work for publication.

Author Response

According to the Authors' answers, all corrections were taken into account, which is not true. The authors only partially took into account the comments contained in the review. Until they are not included, I cannot recommend the work for publication.

  • We would like to thank you again for your time and precious comments that make this manuscript highly qualified in the field of inorganic-polymer composites. We have reconsidered your comments as you gave us for the revision-round 1 and carefully revised the manuscript according to your comments in the details below.
  • We revised the measurement of the density of all composites by using Archimedes method. We have shown the new density and void content of the composites in Table 1, Line 170-171, and rewritten in the main text, Line 157, 158 and 162.
  • We have shown that the density of commercial O-MMT is in the range of 0.2-0.5 g/cm3 in Section 3.1, Line 149-150. The material data sheet of commercial O-MMT which provided by Sigma Aldrich is in this link (https://drive.google.com/file/d/10KcTVIoMjEazgrcOxOSDkq7MksIGe6TP/view?usp=sharing).
  • For DMA results, the main reason for the significant increase in sample stiffness is because the ammonium ion in O-MMT can bond with the hydroxy methyl group in the phenolic resin. Thus, the interfacial adhesion between O-MMT and phenolic matrix are increased, leading to higher stiffness of the polymer composite. We have written this reason in Section 3.3, Line 239-243 and added the Figure 5 of the chemical structure of trimethyl stearyl ammonium modified nanoclay in Line 247-248.
  • The crosslink density analysis part made this manuscript more valuable. The theory can be applied to explain the experiment results perfectly. To our knowledge, this theoretical calculation is well known in the thermosetting polymer study. We found that the crosslink density analysis of the polymer composites corresponds to the work of Ahmadijokani, F., Shojaei, A., Arjmand, M., Alaei, Y., Yan, N. Effect of short carbon fiber on thermal, mechanical and tribological behavior of phenolic-based brake friction materials. Compos. Part B 2019, 168, 98–105; doi:10.1016/j.compositesb.2018.12.038. They showed that the crosslink density value was used to confirm the improvement of thermo-mechanical property of polymer composites. Therefore, we would like to include this part in our manuscript; however, if you do not agree with us, we can exclude it.
